# Forecasting Diabetes Cases Prevented and Cost Savings Associated with Population Increases of Walking in the Greater Toronto and Hamilton Area, Canada

**DOI:** 10.3390/ijerph18158127

**Published:** 2021-07-31

**Authors:** Kathy Kornas, Laura C. Rosella, Ghazal S. Fazli, Gillian L. Booth

**Affiliations:** 1Dalla Lana School of Public Health, University of Toronto, Toronto, ON L5L 1C6, Canada; kathy.kornas@utoronto.ca; 2ICES, Toronto, ON M4N 3M5, Canada; 3Institute for Better Health, Trillium Health Partners, Mississauga, ON L5B 1B8, Canada; 4MAP Centre for Urban Health Solutions, Li Ka Shing Knowledge Institute of St. Michael’s Hospital, Toronto, ON M5B 1W8, Canada; ghazal.fazli@unityhealth.to (G.S.F.); gillian.booth@unityhealth.to (G.L.B.); 5Institute of Health Policy, Management and Evaluation, University of Toronto, Toronto, ON L5L 1C6, Canada; 6Department of Medicine, St. Michael’s Hospital and the University of Toronto, Toronto, ON M5B 1W8, Canada

**Keywords:** population-level, prevention, attributable costs, prediction model, type 2 diabetes

## Abstract

Promoting adequate levels of physical activity in the population is important for diabetes prevention. However, the scale needed to achieve tangible population benefits is unclear. We aimed to estimate the public health impact of increases in walking as a means of diabetes prevention and health care cost savings attributable to diabetes. We applied the validated Diabetes Population Risk Tool (DPoRT) to the 2015/16 Canadian Community Health Survey for adults aged 18–64, living in the Greater Toronto and Hamilton area, Ontario, Canada. DPoRT was used to generate three population-level scenarios involving increases in walking among individuals with low physical activity levels, low daily step counts and high dependency on non-active forms of travel, compared to a baseline scenario (no change in walking rates). We estimated number of diabetes cases prevented and health care costs saved in each scenario compared with the baseline. Each of the three scenarios predicted a considerable reduction in diabetes and related health care cost savings. In order of impact, the largest population benefits were predicted from targeting populations with low physical activity levels, low daily step counts, and non active transport use. Population increases of walking by 25 min each week was predicted to prevent up to 10.4 thousand diabetes cases and generate CAD 74.4 million in health care cost savings in 10 years. Diabetes reductions and cost savings were projected to be higher if increases of 150 min of walking per week could be achieved at the population-level (up to 54.3 thousand diabetes cases prevented and CAD 386.9 million in health care cost savings). Policy, programming, and community designs that achieve modest increases in population walking could translate to meaningful reductions in the diabetes burden and cost savings to the health care system.

## 1. Introduction

Direct health spending on diabetes has increased worldwide due to rising numbers of people living with diabetes, but also as a result of higher year over year medical spending on people with diabetes [1,2]. In Canada, the number of new cases of type 2 diabetes were projected to increase by 2.16 million between 2011 and 2021, which amounts to an estimated CAD 15.36 billion dollars in health care costs [3].

Sedentary lifestyle and physical inactivity levels are increasing world-wide [4] and are among the most important risk factors for developing diabetes [5]. Global physical activity guidelines recommend that adults accumulate 150 min of moderate- to vigorous-intensity physical activity per week, which includes activities such as walking briskly, bicycling, and running [6]. Regular walking of 150 min per week has been shown to reduce the risk of Type 2 diabetes by 30%, as compared to almost no walking [7]. However, the dose–response relationship between physical activity and type 2 diabetes suggests that even minor increases in physical activity (i.e., lower volume and/or intensity) will reduce diabetes risk [8]. In a systematic review which included 28 prospective cohort studies, 2.25 metabolic equivalence of task (MET) h/week (equivalent to 30 min/week of moderate physical activity) was associated with a 7% risk reduction in type 2 diabetes, and further increases of 11.25 MET h/week was associated with a 26% risk reduction [8].

Aspects of the built environment, such as neighbourhood walkability, access to active transport infrastructure, and proximity to green space are key determinants of walking and other forms of physical activity [9,10]. U.S. cities that are more walkable have been shown to have substantially smaller variations in population-level rates of walking, referred to as ‘activity inequality’ [11]. The built environment has also been linked to diabetes risk, for example, people living in highly walkable neighbourhoods have a lower incidence of diabetes than their counterparts living in low walkability neighbourhoods [12]. Additionally, modeling studies examining the effects of active transportation policies that encourage walking have demonstrated health benefits in terms of reductions in diabetes incidence [13,14].

Governments have a role in creating environments, policy, programming and other opportunities that promote active living and physical activity [15,16]. The World Health Organization’s Global Action Plan for the Prevention and Control of Non-Communicable Diseases has proposed policy options around the built environment to increase population levels of physical activity [6]. However, a challenge for decision-makers is identifying appropriate targets for intervention and the scale needed to achieve population impact [17,18]. Population based risk tools are useful for characterizing the distribution of risk in a population and for modeling the population benefit realized from prevention strategies [19].

The objective of this study was to compare scenarios focused on small changes in population walking patterns in a large metropolitan area in Ontario, Canada, and model the impact on diabetes cases prevented and health care cost savings attributable to diabetes over a 10-year period. To do so, we applied a validated population based risk tool to estimate the impact of prevention scenarios that focus on walking. This study demonstrates how the tool can be applied for modeling the impact that changes in baseline risk factors will have on future diabetes incidence.

## 2. Materials and Methods

### 2.1. Context and Setting

This study focused on the population living in the Greater Toronto and Hamilton area in the province of Ontario, one of the largest and fastest growing metropolitan areas in North America, with a population of 7.5 million people in 2016 [20]. This study was approved by the University of Toronto Health Sciences Research Ethics Board.

### 2.2. Study Population

Our study population included participants 20 to 64 years old in the 2015 and 2016 cycles of the Canadian Community Health Survey (CCHS). The detailed survey methodology of the CCHS is described elsewhere [21]. Briefly, the CCHS is a cross-sectional population-based survey that collects self-reported health information from a representative sample of Canadians aged 12 years and older who are living in private dwellings (~98% of the Canadian population). The sample frame for the adult population (18 years and older) uses an area frame of a selection of households drawn from Statistics Canada’s Labour Force Survey; the area frame uses a two-stage stratified cluster design. The CCHS sampling frame excludes individuals living in long-term care institutions, on Indigenous Reserves, Canadian Forces Bases, and some remote areas. For the 2015 and 2016 CCHS, 26,388 households in Ontario were in scope for the survey, out of which 15,759 individuals responded (response rate of 59.7%). The CCHS is conducted using computer-assisted personal or telephone interviewing.

### 2.3. Baseline Variables

Baseline variables were captured from the CCHS and were categorized according to variable categorizations used for the Diabetes Population Risk Tool (DPoRT). Sociodemographic characteristics included age category (20–44 and 45–64 years old), sex, and household income quintile. Body mass index was calculated with self-reported weight and height using a standard formula (weight in kilograms divided by height in meters squared), and was categorized according to normal weight (<25.0 kg/m^2^), overweight (25.0–29.9 kg/m^2^ ) and obese (≥ 30.0 kg/m^2^).

Physical activity was assessed based on activities reported for the past 7 days, and categorized according to sedentary (0 MET min/week), somewhat active (1–449 MET min/week), moderately active (450–899 MET min/week), and active (≥900 MET min/week). To do so, respondents reported the amount of minutes in which they engaged in three activity categories: active transportation (used active ways like walking or cycling to get to places such as work, school, bus stop, shopping centre or to visit friends), leisure-time activities (organized or non-organized that lasted a minimum of 10 continuous minutes, such as exercise, swimming, and all team sports), and other activities that lasted a minimum of 10 continuous minutes (carrying heavy loads, shovelling, and household chores). Of the total time spent on these activities, respondents reported the number of minutes that were of vigorous intensity. Time spent performing physical activities was multiplied by the MET value assigned to each activity category (METS x minutes/week) to derive the overall level of physical activity undertaken. Active transportation, leisure time and other activities were considered to be of moderate level intensity requiring 3 METS/minute, while vigorous intensity activities were considered to require 6 METS/minute [22].

Individuals were further categorized based on the baseline amount of physical activity derived from active forms of transportation in the last 7 days (0, 1–75, 76–149, ≥150 min). Based on a previous study, moderate intensity walking, as defined as 3 METs, was deemed to be equivalent to 100 steps per minute [23]. Therefore, we derived the average daily step counts from active transportation by computing from the total number of minutes reported using active forms of transportation and the following equation: daily step count = (weekly active transportation minutes x100)/7.

### 2.4. Outcome Measures and Analysis

The study used the Diabetes Population Risk Tool (DPoRT), a validated algorithm, to estimate the average risk of diabetes at baseline and to model the number of diabetes cases prevented and health care costs saved as a result of changes in population walking patterns (our test scenarios).

### 2.5. Estimating 10-Year Diabetes Risk

DPoRT is a population-based risk tool that estimates the ten-year incidence of physician diagnosed type 2 diabetes among adults 20 years and older who are currently without diabetes. The tool can be applied to quantify impact that changes in risk factors will have on future diabetes incidence. DPoRT predicts the probability of developing diabetes using a statistical model based on the Weibull survival distribution. DPoRT is validated to calculate up to 10-year diabetes risk in any population-based data that contains self-reported risk factor information on age, sex, body mass index, ethnicity, education, immigrant status, prior diagnosis of hypertension, prior heart disease, household income quintile, and smoking. The original risk algorithm was developed based on a cohort of 19,861 individuals without diabetes followed between 1996 and 2005, and was validated in two external cohorts in the provinces of Ontario (*n* = 26,465) and Manitoba (*n* = 9899), as well as across ethnic groups [24,25,26]. The cohorts linked baseline risk factors to a validated population-based diabetes registry to ascertain diabetes diagnosis during follow-up. The algorithm coefficients were updated with more recent data from an Ontario cohort (*n* = 69,606), with follow-up until 2011 [25]. The updated DPoRT model has demonstrated high overall predictive performance, good discrimination (C= 0.77) and calibration [25]. Full details of development and validation can be found from a previous study [24,25]. The algorithm formula and risk factor coefficients can be found in Appendix A.

DPoRT estimated each individual’s 10-year risk of developing diabetes. To do this, the DPoRT algorithm was calculated using centered adjusted coefficient values in order to account for differences between populations and adjust for baseline risk factors. Therefore, we first applied DPoRT risk equations to the CCHS sample representing the Ontario provincial population, excluding those less than 20 years old, those with a history of diabetes, and those with missing risk factor information required for DPoRT (3.9%). After determining individuals’ baseline diabetes risk, we subsequently excluded individuals who did not meet the criteria of our study population; specifically, individuals over the age of 64 and those who were not a resident of the Greater Toronto and Hamilton area. Diabetes risk estimates were then averaged across all respondents of the study population to determine the population-level risk of diabetes at baseline and after each scenario described below. The number of new (incident) cases of diabetes was estimated by multiplying the average risk by the population size.

### 2.6. Estimating Health Care Costs Attributable to Diabetes

We estimated the direct health care costs attributable to diabetes using a previously developed cost calculator that multiplied annual diabetes incidence predictions from DPoRT by estimates of excess health care spending on diabetes patients, obtained from a propensity-matched cohort study in Ontario [3]; full methodological details are described elsewhere [27]. Briefly, this study used the Ontario Diabetes Database to identify new cases of physician-diagnosed diabetes from 1 April 2004 to 31 March 2012. Three control subjects without diabetes were matched to each person with diabetes; subjects were matched on index date (±30 days), age (±90 days) and the logit of the propensity score. This score was the predicted probability of developing or not developing diabetes, calculated from a logistic regression consisting of age, rurality, comorbidity, geographic location and neighbourhood income quintile. A person-centered costing approach was applied to estimate direct health care costs of diabetes over the eight years of follow-up. Costs covered health care encounters accrued through Ontario’s single payer government insurer, including inpatient hospitalizations, emergency department visits, physician services, same-day surgeries, prescriptions, rehabilitation, complex continuing care, mental health inpatient stays, long term care, and home care services. The study determined attributable costs as the annual difference in cost between those with and without diabetes, reported in 2012 Canadian dollars. The cost calculator multiplied the number of estimated diabetes cases for each year by the corresponding per-patient annual excess cost (determined from the previous study [27]). Calculations were sex-specific and took into account time since diabetes diagnosis and annual mortality rates. Mortality rates were specific to year of follow-up. Since individual costing estimates in the original analysis used eight years of follow-up, the cost calculator assumed attributable costs in years 9 and 10 after diagnosis to be the same monetary value as in year 8. Full details on the development of the cost calculator are described elsewhere [3].

### 2.7. Modeling the Effects of Increased Walking

We developed three scenarios focused on changes in population walking patterns and their estimated impact on diabetes risk. Each scenario modelled the effect of increases in different walking durations for a target group within our study population (i.e., individuals who have low levels of walking at baseline who would be willing to change their walking behaviour). A description of each scenario, including the target population and effects modelled in each scenario are summarized in Table 1.

We defined three different population target groups for modelling: individuals with low overall levels of physical activity (scenario 1), individuals with low daily step counts from active transportation (scenario 2), and individuals who did not use active forms of travel in the past week (scenario 3). For scenario 1, we defined low levels of physical activity by categorizing respondents’ weekly physical activity levels into quartiles, and assigning those in the lowest two quartiles of physical activity to the target group. The target group for scenarios 2 and 3 were individuals who had low daily step counts from active transportation (between 0 and 999 steps) and those who did not undertake any forms of active transportation (0 steps) in the past week. We aimed to define target groups that could benefit from increases in walking, thus, in each scenario, the target group coverage excluded individuals who reported high physical activity levels at baseline (quartiles 3 & 4).

For each target group, we applied a relative risk reduction to the DPoRT-estimated diabetes risk to quantify the expected impact of increases in weekly walking from baseline levels. We chose to model an upper limit of 150 min of walking per week to correspond to weekly physical activity recommendations for adults. In our modeling, we included lower thresholds of 25, 50, 75, and 100 min of walking per week to examine the potential benefit of a population undertaking an increase in some walking that is lower than the weekly physical activity recommendation, but which could potentially be encouraged through public health intervention. Relative risk reduction values were obtained from a published meta-analysis of prospective cohort studies, which reported a 26% relative risk reduction for type 2 diabetes incidence among those who achieved 11.25 MET hour/week of leisure-time physical activity, equivalent to 150 min per week of moderate activity [8]. To determine the relative risk reduction associated with 25, 50, 75, and 100 min of walking per week, we adjusted the relative risk reduction value observed in the meta-analysis by linearizing the risk reduction on the natural log scale.

All analyses were weighted using sampling weights provided by Statistics Canada to adjust for the complex sampling design of the CCHS and to produce estimates reflecting the population living in the Greater Toronto and Hamilton area. The weights correspond to the number of persons in the population that are represented by the respondent. The weights were determined based on an initial weight provided by the Labour Force Survey design, and the weighting strategy included sub-cluster adjustment, stabilization, and adjustments for household and person-level nonresponse. The population distribution of sociodemographic and behavioural characteristics of the study sample was examined overall and by active transportation use. We compared the estimated 10-year diabetes cases and costs corresponding to each scenario against the baseline. We defined population benefit as the absolute risk reduction in diabetes risk (absolute difference in diabetes risk at baseline and diabetes risk after the scenario) and number of diabetes cases prevented with each scenario (absolute difference in diabetes cases at baseline and diabetes cases after the scenario). Health care cost savings were calculated as the absolute difference in costs attributable to diabetes at baseline and costs estimated after the scenario. Data analyses were performed in SAS version 9.4 (SAS Institute, Cary, NC, USA). Calculations using the DPoRT cost calculator were performed in Excel 2010 (Microsoft).

## 3. Results

Overall, there were 32,928 individuals who participated in the CCHS between 2015 and 2016 and were living in the province of Ontario at the time of their interview date. Using DPoRT equations, baseline diabetes risk was determined for all individuals residing in the province, excluding 3262 respondents who were less than 20 years old, 2974 respondents with a prior diagnosis of diabetes, and 3075 respondents with missing risk factor information required for DPoRT. Of the remaining 23,617 individuals, we further excluded individuals who did not meet our eligibility criteria based on their age (20–64 years) and location of residence (Figure 1). As a result, a total of 5160 CCHS respondents were retained for the analysis, representing a population of 3,857,703 adults aged 20–64 years who were living in the Greater Toronto and Hamilton area.

Table 2 shows the weighted characteristics of individuals in our cohort based on the mode of transportation they reported using in the past 7 days. At the time of their interview, those who used active forms of transportation (walking, cycling or public transit) were younger, had lower BMI levels, and higher likelihood of being physically active than those reporting not using active forms of travel. The distribution of household income between those using active and non-active transportation was similar.

The baseline predicted incidence of diabetes and health care costs incurred for new diabetes cases between 2015/16 to 2025/26 is provided in Table 3. Among all members of our study population, the predicted 10-year incidence was 8.9%, translating to 345 thousand new diabetes cases and an estimated CAD 2.45 billion in total health care costs attributable to diabetes. The distribution of new diabetes cases over the 10-year period varied according to the baseline physical activity and active transportation use reported by members of the cohort (Figure 2). The 10-year diabetes incidence was estimated to be 7.8% among individuals in the population who were physically active compared to 10.3% among those who were sedentary. Similarly, diabetes risk decreased with greater engagement in active transportation, from 10.1% among those with 0 weekly minutes of active transport time to 7.5% among those with ≥150 weekly minutes. Of the total new diabetes cases estimated, about half (180 thousand cases) were predicted to occur among those who reported not using an active mode of transportation in the past week.

Scenarios in which members of the cohort with low physical activity levels, low step counts, or no use of active transportation were assumed to undertake more minutes of walking each week are presented in Table 3. The first series of scenarios estimated the impact of increased walking among members of the cohort with the lowest physical activity levels (quartiles 1 and 2). These models predicted that an additional 25 min per week of walking would result in 10.4 thousand fewer diabetes cases over 10 years among the least physical active groups, amounting to CAD 74.4 million in health care cost savings (Table 3). If walking time increased by an additional 150 min each week, the models estimated that there would be 54.3 thousand fewer diabetes cases and CAD 386.9 million in health care cost savings.

Similar findings were noted in regard to the second and third set of scenarios. If the distribution of step counts were narrowed so that those with less than 1000 steps per day walked an additional 25 min per week (equivalent to an additional 360 steps each day) there would be an estimated 8.2 thousand fewer cases of diabetes and CAD 58.4 million lower health care costs (Table 3). Increasing walking by an additional 150 min per week (equivalent to an increase in daily step counts of 2100) was estimated to result in 42.7 thousand fewer diabetes cases, resulting in potential savings of CAD 303.7 million in health care costs in ten years. Similarly, DPoRT predicted 6.6 thousand and 34.4 thousand fewer diabetes cases among non-active transport users in our cohort if they increased their weekly walking time by 25 and 150 min, respectively, and corresponding health care savings estimated to be CAD 47.1 to 244.8 million.

## 4. Discussion

A central tenet of population approaches to diabetes prevention is the need to shift the distribution of diabetes risk [17]. Our research suggests that this is possible with only small changes in a key risk factor, such as physical activity. In this study, we estimated that population-level increases in walking of as little as 25 min per week, could translate to meaningful reductions in the diabetes burden and health care cost savings within 10 years.

The results highlight the importance of physical activity as a target for diabetes prevention, and the potential for policy interventions that promote walking to reduce the burden and costs of diabetes. Other studies have demonstrated the benefits of walking on blood pressure, serum lipid levels, and weight loss [28]. Physical activity targets have been established around the world, for example, member states of the World Health Organization have agreed to a 10% relative reduction in the prevalence of insufficient physical activity, defined as less than 150 min of moderate-intensity activity per week, by 2025 [6]. Although countries have shown slow progress towards meeting these physical activity targets [4], the findings highlight that a population-wide approach would require only a modest increase of 25 min in weekly walking to achieve important reductions in diabetes and cost savings. Relevant policy options that can potentially promote increases in walking could include investments in active transport and public transit infrastructure [29]. Previous studies have shown that increased access to public transit and increased proportions of the population who walk to work can promote attainment of physical activity guidelines [29,30].

The findings characterize baseline risk of developing diabetes in a jurisdiction and quantify the trade-offs of population and targeted prevention approaches that focus on increases in walking; this approach is informative for appropriately planning diabetes prevention strategies. Specifically, our modelling scenarios showed that, in this jurisdiction, targeting individuals who use non-active forms of travel, such as cars, could result in meaningful diabetes reductions. However, a single strategy, particularly those that elicit small changes at the population level will be insufficient at curbing the rise in the diabetes burden. The results suggest that achieving a sizable reduction in diabetes cases will require a mix of multi-modal strategies that have a broad reach in populations who are at risk, including those who are physically inactive. Policy options for promoting changes in physical activity and walking patterns are diverse, but underlying mechanisms include actions that change the built environment and community design [31].

This study’s findings showing diabetes reductions from increases in walking from active travel are comparable to those of other studies that have modeled the effects of increases in walking through transportation policy initiatives. A modeling study examining the effect of a municipal transportation policy to increase active transportation in Ottawa, Ontario found that 1620 incident cases of diabetes could be prevented in 10 years by increases in public transit use [14] Another modeling study examining the impacts of high walking and cycling transport scenarios in the UK urban environment found a reduction of 7.2% in the diabetes burden by 2030 [13].

The results should be interpreted with consideration of the study limitations. First, the 10-year diabetes risk estimates are based on the distribution of population risk factors in the baseline year, 2015/16. The estimates do not account for population growth or changes in population demographics over the forecast period. Second, the use of self-reported measures on physical activity are subject to social desirability and recall bias [32]; although, the use of a 7-day look-back window may have reduced inaccuracies in reporting. It is important to recognize that measuring health behaviours at the population-level is currently difficult to achieve any other way. Third, the scenarios do not account for increases in walking by adults greater than 65 years old. However, we assumed that interventions related to active transportation (e.g., time spent walking to and from public transit) would potentially have greater relevance and uptake by the school and working population aged less than 65, and therefore we took a conservative approach by excluding adults over 64 years old from the modeling scenarios. Nonetheless, it is likely that this resulted in an underestimate of diabetes cases prevented and cost savings associated with these scenarios. Finally, projected estimates of cost savings attributable to diabetes do not account for inflation, changes in health care spending for treating diabetes over the forecast period, productivity losses, or informal care costs. It is likely that the absolute cost savings projected for each scenario are an underestimate, however, our primary purpose was to compare the relative cost savings between the different intervention scenarios. The current study focused on the impact of increases in walking in reducing diabetes risk at the population-level to inform future directions for diabetes prevention. Future research can apply DPoRT to examine the impact of changes to other modifiable baseline risk factors, such as diet and obesity.

## 5. Conclusions

Using a validated population risk tool, we demonstrated that a modest shift in the population’s walking behavior in our jurisdiction could result in meaningful reductions in diabetes and health care cost savings. The findings emphasize that policy, programming, and community designs that encourage physical activity and active living at the population-level are a meaningful target for diabetes prevention efforts.

## Figures and Tables

**Figure 1 ijerph-18-08127-f001:**
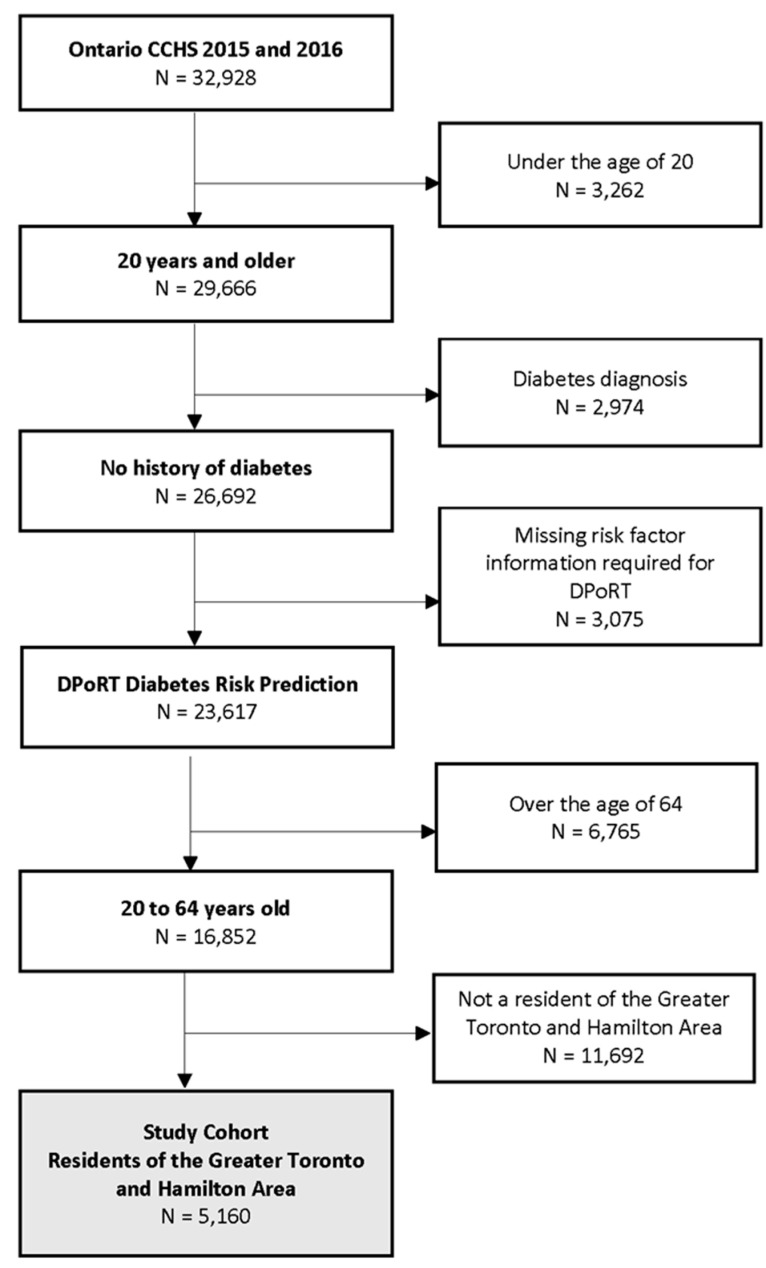
Cohort inclusion and exclusion criteria.

**Figure 2 ijerph-18-08127-f002:**
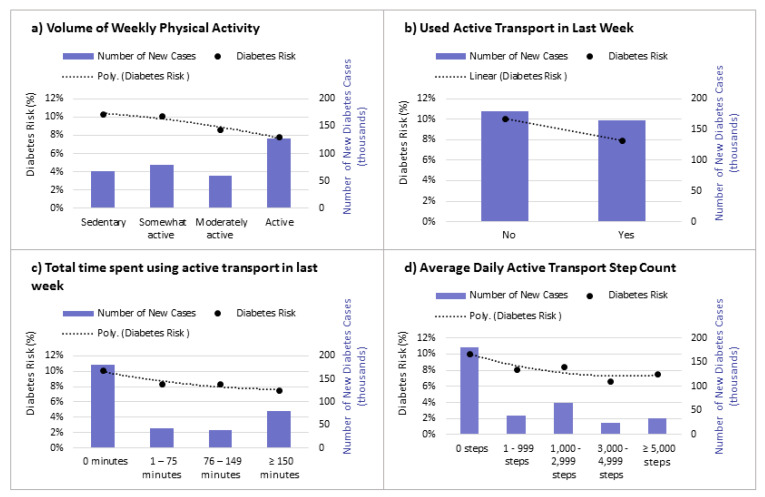
10-year DPoRT estimated diabetes risk and new cases (2015/16–2025/26) according to: (**a**) weekly physical activity level; (**b**) active transport use in last week; (**c**) total minutes spent using active transport in past week; (**d**) average daily active transport step count.

**Table 1 ijerph-18-08127-t001:** Summary of population targets and effects used to define each scenario modelling the impact of increases in walking on diabetes.

	Scenarios Modelling Increases in Walking		
Scenario	1	2	3
Description	Individuals with low weekly physical activity levels increase their walking time from baseline levels.	Distribution of step counts are narrowed by increases in walking among those with low daily step counts from active transportation.	Individuals who did not report using active transportation increase their walking time from baseline levels.
Population Target Group	All individuals with low weekly physical activity levels (quartiles 1 and 2).	All individuals with <1000 steps per day.	All individuals who reported not using active transportation in the past week.
Target Group Coverage	Effects were not applied to subjects who reported high weekly physical activity levels (quartiles 3 & 4)		
Effects Relative risk reduction (RR) associated with additional walking per week from baseline levels [7]	25 min = 5% RR 50 min = 10% RR 75 min = 14% RR 100 min = 18% RR 150 min = 26% RR		

**Table 2 ijerph-18-08127-t002:** Weighted distribution of baseline characteristics for the study population.

Characteristic	Overall	Used Active Transport ^b^	Did Not Use Active Transport ^b^
Represented Population ^a^	*n* = 5160 3,857,703	*n* = 2563 2,075,863	*n* = 2594 1,781,840
Sex (male)	47.9	45.8	50.2
Age at interview			
20–44	57.6	62.4	52.1
45–64	42.4	37.6	47.8
Household income			
Q1 (lowest)	19.3	19.9	18.5
Q2	19.5	20.9	17.9
Q3	19.8	18.8	21.0
Q4	21.3	20.1	22.8
Q5 (highest)	20.1	20.2	19.9
Body Mass Index			
BMI < 25.0	49.0	53.3	44.1
BMI 25.0–29.9	32.7	30.4	35.3
BMI ≥ 30.0	14.5	12.6	16.7
Weekly physical activity in last 7 days			
Sedentary	17.5	0.0	37.8
Somewhat active	21.0	22.4	19.3
Moderately active	18.6	22.5	14.0
Active	42.9	55.0	28.9
Total active transportation minutes in last 7 days			
0 min	46.2	0.0	0.0
1–75 min	13.5	25.1	0.0
76–149 min	12.1	22.6	0.0
≥150 min	27.7	51.6	0.0
Average daily active transportation step count			
0 steps	46.2	0.0	100.0
1–999 steps	12.9	24.0	0.0
1000–2999 steps	19.9	37.0	0.0
3000–4999 steps	9.3	17.3	0.0
≥5000 steps	11.2	20.8	0.0

^a^ Represented population estimated using the Canadian Community Health Survey sampling weights. Percentages may not total 100% because of missing observations and rounding; ^b^ In the last 7 days, respondent used/did not use active ways like walking or cycling to get to places such as work, school, the bus stop, the shopping centre or to visit friends.

**Table 3 ijerph-18-08127-t003:** Ten-year diabetes incidence rate, number of new diabetes cases, and costs attributable to diabetes by scenario (2015/16–2025/26).

**SCENARIO 1: Individuals with Low Weekly Physical Activity Levels (Quartiles 1 & 2) Increase Their Walking Time from Baseline Levels.**						
Extra minutes spent walking per week ^a^	**Baseline**	**+25 min**	**+50 min**	**+75 min**	**+100 min**	**+150 min**
10-year diabetes risk (%)	8.9	8.7	8.4	8.2	8.0	7.5
Number of new diabetes cases (thousands)	345.0	334.6	324.1	315.8	307.4	290.7
Health care costs attributable to diabetes (CAD, billions) ^b^	2.45	2.37	2.30	2.24	2.18	2.06
Absolute risk reduction from baseline (%)	-	0.2	0.5	0.7	0.9	1.4
Number of diabetes cases prevented (thousands)	-	10.4	20.9	29.2	37.6	54.3
Health care cost savings (CAD, millions) ^a^	-	74.4	148.8	208.4	267.9	386.9
**SCENARIO 2: Individuals with <1000 steps per day increase their walking time from baseline levels.**						
Extra minutes spent walking per week ^a^	**Baseline**	**+25 min**	**+50 min**	**+75 min**	**+100 min**	**+150 min**
10-year diabetes risk (%)	8.9	8.7	8.5	8.3	8.2	7.8
Number of new diabetes cases (thousands)	345.0	336.8	328.6	322.0	315.5	302.3
Health care costs attributable to diabetes (CAD, billions) ^b^	2.45	2.39	2.33	2.29	2.24	2.15
Absolute risk reduction from baseline (%)	-	0.2	0.4	0.6	0.7	1.1
Number of diabetes cases prevented (thousands)	-	8.2	16.4	23.0	29.5	42.7
Health care cost savings (CAD, millions) ^b^	-	58.4	116.8	163.5	210.2	303.7
**SCENARIO 3: Individuals who reported not using active transportation in the past week increase their walking time from baseline levels.**						
Extra minutes spent walking per week ^a^	**Baseline**	**+25 min**	**+50 min**	**+75 min**	**+100 min**	**+150 min**
10-year diabetes risk (%)	8.9	8.8	8.6	8.5	8.3	8.0
Number of new diabetes cases (thousands)	345.0	338.4	331.8	326.5	321.2	310.6
Health care costs attributable to diabetes (CAD, billions) ^b^	2.45	2.40	2.35	2.32	2.28	2.20
Absolute risk reduction from baseline (%)	-	0.1	0.3	0.4	0.6	0.9
Number of diabetes cases prevented (thousands)	-	6.6	13.2	18.5	23.8	34.4
Health care cost savings (CAD, millions) ^b^	-	47.1	94.1	131.8	169.5	244.8

^a^ The relative risks associated with minutes spent walking per week were taken from a systematic review (Smith et al., 2016) [8]. ^b^ Costs are reported in Canadian dollars for 2012.

## Data Availability

Restrictions apply to the availability of these data. Data was obtained from Statistics Canada and are available at https://www.statcan.gc.ca/eng/microdata (accessed on 7 January 2019) with the permission of Statistics Canada.

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
