# Peer review of "Forecasting Diabetes Cases Prevented and Cost Savings Associated with Population Increases of Walking in the Greater Toronto and Hamilton Area, Canada"

_ijerph, 2021, doi:10.3390/ijerph18158127_

Round 1

Reviewer 1 Report

Thanks for recommending me as a reviewer. This paper was aim to estimate the public health impact of increases in walking as a means of diabetes prevention and health care cost savings attributable to diabetes. If the authors complete revisions, the quality of the study will be further improved.

  1. The introduction section is well written. If the author describes more specifically public health impact of increases in walking as a means of diabetes prevention and health care cost savings attributable to diabetes in the introduction section, it can help readers understand.

2. Authors should be more specific about the Canadian Community Health Survey. For example, authors need to be more specific about sampling, data properties, collection method, missing valuesetc. in the Methods section.

3. line 112-120: If the authors describe the Diabetes Population Risk Tool (DPoRT) in more detail in the Methods section, it may help readers to understand. 

4. line 193-200: How were the weights calculated in this study?

5. In Table 1, "Represented population" seems to be unnecessary. In my opinion, it would be better to present only the weighted average.

6. In Table 3, what is the criteria for classifying “Extra minutes spent walking per week”? For example, +25 minutes + 50 minutes + 75 minutes + 100 minutes + 150 minutes are classified according to what criteria?

7. The discussion section is well written.

Reviewer 2 Report

First of all, I would like to thank the editor and the authors of this paper for giving me the opportunity to read the paper entitled "Forecasting Diabetes Cases Prevented and Cost Savings Associated with Population Increases of Walking in the Greater Toronto and Hamilton Area, Canada". The paper is very well-written and provides new evidence on the positive effect of initiating healthy behaviours. However, I do still have some concerns regaridng the current status of the paper, which I detail below.

ABSTRACT

  • I would add some information on the effect of the other scenarios, not only the 25-min-walking.

INTRODUCTION

  • What is the definition or which activities are considered as moderate- to vigorous-intensity physical activity? Maybe you can repeat the definition included in lines 100-103.
  • Lines 59-61: it seems that you aim to identify population targets. Please, provide a better link with the purpose of your study and provide a brief description of your contribution to the existing literature.

METHODS

  • Why is the sample limited to people aged 20 to 64 years old? Type 2 diabetes mellitus represents 90-95% of all the diabetes mellitus. Type 2 diabetes mellitus prevalence on adults aged 65 yeas old and above is 25% and one in every four older adults (>65 years old) has type 2 diabetes mellitus. If you exclude these individuals, the effect of physical activity might be underestimated.
  • Line 136: why did you exclude individuals aged 20 years old and above?

DISCUSSION

  • The Discussion section lack a more in-depth comparison with the existing literature. Please, add.
  • In the limitations you could add something about the exclusion of productivity losses and informal care costs, which also pose a great burden in economic terms for people with diabetes.
  • Add future lines of research and policy implications.

GENERAL COMMENT

Revise the writing as I have identified some typos (for example, first line of the introduction: due to (instead of due))

Reviewer 3 Report

This manuscript provides us a comparison about scenarios focused on small changes in population walking patterns in a large metropolitan area in Ontario, Canada, and models their impact on diabetes cases and health care cost savings attributable to diabetes over a 10-year period. The topic is very relevant and the study design and procedure are very clear. In my opinion, this is a good paper which covers an interesting and very useful topic with impact in practice. However, the authors seem to have not considered that there are others risk factors modifiable in diabetes, such as diet, so they should consider this limitation and add it to the text.

I would like to make several suggestions for revision:

Introduction

Line 43- – please provide a reference in the end of “…activity (i.e., lower volume and/or intensity) will reduce diabetes risk.”

Material and methods

Line 83- “ (20-44 and 45-64 years old)”-Why the authors chose these age categories? The authors need to clarify these aspects in the methodology part for the readers to understand this choice.

Line 120, Line 150, line 162- …”described elsewhere”. Throughout the article the authors wrote this type of sentence several times and forward to the corresponding reference. However, for the readers to understand the methodology, I recommend that authors should make a brief description and not only forward to the corresponding reference.

Line 124-“ groups.21,23” The authors should adequately format the references in the text.

Round 2

Reviewer 1 Report

The authors faithfully completed the revision.

Reviewer 2 Report

No further comments are made to the current version of the manuscript. I would like to congratulate the authors on he hard work and effort placed to improve the quality of the manuscript.

Reviewer 3 Report

The authors clearly improved the manuscript, therefore, in my opinion, it is now susceptible for publication